# Validation of the Multidimensional Sociosexual Orientation Inventory (SOI-M) in the Chilean population

Oriana Figueroa[1,2], Pablo Polo[2*], José Antonio Muñoz-Reyes[2], Rodrigo Ferrer[3], Gabriela Fajardo[4], Daniel Torrico-Bazoberry[2]

**1** Facultad de Salud y Ciencias Sociales, Escuela de Psicología, Universidad de las Américas, Santiago, Chile, **2** Laboratorio de Comportamiento Animal y Humano, Centro de Investigación en Complejidad Social, Facultad de Gobierno, Universidad del Desarrollo, Santiago, Chile, **3** Facultad de Ciencias Sociales, Universidad de Tarapacá, Arica, Chile, **4** Facultad de Administración y Economía, Universidad de Santiago de Chile, Santiago, Chile

\* p.polo@udd.cl

## Abstract

We used the SOI-M Multidimensional Sociosexual Orientation Inventory developed by Jackson and Kirkpatrick for validation in the Chilean population. This 3-dimensional instrument measures sociosexuality in terms of short-term mating orientation, long-term mating orientation, and sociosexual behavior. This inventory allows us to capture the deployment of reproductive strategies in men and women simultaneously (that is, in the short and long-term). We tested the psychometric properties of the instrument on a final sample of 865 subjects (247 women and 616 men) aged between 18 and 55 years old (M = 23.56; SD = 5.52). First, with a subsample (N = 172), we performed an EFA to establish dimensionality. Then, with the remaining sample (N = 693) an ESEM and CFA. We performed an ESEM for confirmatory analysis, analyzed invariance by sex and relations with other variables. We verified the existence of three factors, maintaining the 20 original items. Then, we eliminated the redundant or cross-factor loading items, thus obtaining a reduced version of 15 items and three factors. We tested and verified the convergent and divergent validity of the instrument. Also, we tested invariance by sex, establishing that the SOI-M behaves invariantly. The SOI-M is a valid and reliable instrument to measure multidimensional sociosexuality in the Chilean population.

## Introduction

Sociosexuality can be conceptualized as the individual predisposition to engage in sexual relationships without commitment [1]. Thus, it is considered a relevant construct to study the variability of human mating strategies, both in attitudinal and behavioral terms. This construct, developed in the 1990s by Simpson and Gangestad

**Data availability statement:** Data for the study Validation of the Multidimensional Sociosexual Orientation Inventory in the Chilean population are available in https://osf.io/dcvjz/?view_only=e724cbbda9f24755b5799ab52dd24cd9.

**Funding:** This work was funded by FONDECYT postdoctorado project (3220233) from Agencia Nacional de Investigación y Desarrollo, ANID, Chilean Government to OF. The funders had no role in study design, data collection and analysis, decision to publish, or preparation of the manuscript.

**Competing interests:** The authors have declared that no competing interests exist.

[1], builds on previous work by Kinsey and colleagues on human sexuality and their findings on variability in phenomena such as promiscuity and sexual permissiveness [2,3]. Kinsey's work was groundbreaking and changed how the expression of human sexuality was understood. First, he was critical about previous studies, since they were almost exclusively developed from a medical perspective, with methodological and moral biases [1,4]. In addition, he established that sexual orientation moves on a continuum between exclusively heterosexual and exclusively homosexual orientation [4]. Finally, he showed that there is a high variability in a wide array of traits that they conceptualized as sociosexual attitudes and behaviors [2,3]. These attitudes and behaviors include the number of sex partners, attitudes towards sex without commitment, and expectations regarding their sexual partners in the future, among others. Subsequent studies showed that many of these traits covaried, which led other authors to postulate that these traits could constitute a single source of individual variation that they called sociosexual orientation or sociosexuality [1]. From here arises the need to develop a psychometric tool that can measure the phenomenon of human sociosexuality to be used on different perspectives and disciplines.

The first version of this questionnaire was the Sociosexual Orientation Inventory (SOI), which consisted of 7 items that measure the willingness to establish sexual relationships without prior commitment as well as questions about the number of sexual partners in the past [1]. Individuals who score high on the SOI would have a more unrestricted orientation characterized by no need of commitment before having sex, a larger number of sexual partners in the last year and during lifetime, a higher frequency of fantasies about having sex with uncommitted partners, and a higher number of expected future sexual partners. In contrast, people who score low on the SOI would have a more restricted orientation characterized by the need of emotional closeness with a romantic partner before having sex and fewer sexual partners during the last year and in life [1]. Consequently, individuals could be classified according to their position in a single bipolar continuum from more restricted to more unrestricted individuals. This first questionnaire, despite its limitations, helped to identify some of the sources of the variability of mating behavior and strategies in humans. More concretely, studies employing the SOI showed that the variability on sociosexual attitudes and behaviors would be partly explained by sexual differences, where men would show a more unrestricted sociosexuality compared to women (e.g., [5,6]). This is because women must invest more time and energy in obligatory parenting, which would make them more selective when choosing a sexual partner, whereas men would increase their fitness by having more sexual partners throughout their lives [1,7]. In addition, this trend has been established in intercultural studies (for a classic example, see [8]). In a study by Schmitt, this instrument was tested in 14 countries and 26 languages, demonstrating its evidence of validity for use in different cultures [6]. In this work, it was established that, almost universally, men would show a greater tendency towards short-term relationships, greater fantasies with multiple partners and less discrimination towards potential partners, especially in contexts of promiscuous mating [6]. However, this approach proposes a one-dimensional vision of the phenomenon and does not recognize that humans might be motivated for

pursuing both strategies –short and long-term mating—simultaneously [9–12]. In addition, this instrument has limitations due to its low level of internal consistency reported in some studies and because it has a structure that does not fit the data well [13,14], which jeopardizes the comprehensive understanding of the phenomenon.

Penke & Asendorpf developed a revised version to address the statistical limitations mentioned above [15]. One of the biggest problems of Simpson and Gangestad's instrument [1] was its one-dimensionality and, therefore, its factorial structure [13]. In this new, revised version of the SOI (SOI-R), the authors proposed a three-factor structure divided into past behavioral experiences, attitudes toward uncommitted sex and sociosexual desires with a total of 9-items [15]. This three-factor structure (behavior, attitude, and desire) is relevant because these dimensions, despite being related, are not completely dependent and refer to different phenomena. Attitudes allude to the positive or negative predisposition toward short-term relationships, behavior refers to the execution of those positive or negative attitudes, and desire refers to a motivational state characterized by a high sexual interest toward potential uncommitted mates [15]. In this sense, an individual may have a very favorable attitude toward sex without commitment, but whether this individual has a high number of sexual partners will depend on both physical and contextual characteristics that will determine success or failure in reproductive terms [16]. This scale has been tested in different countries such as Brazil, Spain, Colombia, Italy and Portugal, obtaining good internal consistency and evidence of validity [17–21]. However, the SOI-R still does not totally incorporate the complexity of reproductive strategies and their variability since it does not include attitudes toward long-term mating as a separate dimension from attitudes toward short-term mating. This can be problematic when interpreting sociosexual attitudes because it neglects the simultaneity of short- and long-term mating strategies and hampers the study of individuals who deploy a dual mating strategy [10,22].

A year before the publication of the SOI-R, Jackson & Kirkpatrick developed an instrument that addresses the problem of unidimensionality in sociosexual attitudes [12]. They developed a three-factor instrument comprising a two-dimensional model for sociosexual attitudes and a one-dimensional model for sociosexual behavior. In this sense, sociosexual attitudes were divided into short-term mating orientation, a dimension that measures the willingness to be involved in a sexual relationship without commitment, being like the first SOI, and long-term mating orientation, a dimension that measures attitudes toward long-term committed relationships independently of short-term attitudes. With this structure, Jackson & Kirkpatrick [12] intended to capture the pluralism in mating behavior reported in humans [9,11]. Consequently, this conceptualization of sociosexuality is more in accordance with the strategic pluralism hypothesis [11], which proposes that despite that men and women would face different trade-offs when deploying their reproductive strategies both have psychological adaptations for pursuing a short and long-term mating strategy in a conditional rather than alternative way allowing them to sequentially or simultaneously pursue these strategies (e.g., [9,11]). This new instrument, the Multidimensional Sociosexual Orientation Inventory (SOI-M), was initially conceived as a 25-items questionnaire, which included items from the original SOI [1], 5 items from the scale of interest in non-committal sex [16], and a question about the number of sexual partners in life. In addition, 9 items were included for the construction of the long-term dimension, and three items were added to the short-term dimension to better reflect women's attitudes associated with this dimension. In short, the first version of SOI-M consisted of 10 items for short-term mating orientation, 10 items for long-term mating orientation, and 5 items for sociosexual behavior. After factor analysis, three items from the long-term mating orientation and two from the sociosexual behavior were discarded because they did not load in any factor. The Cronbach's α values for the final instrument were.95 (short-term mating orientation),.88 (long-term mating orientation) and.83 (previous sexual behavior). Studies employing the SOI-M evidence that are some variables such as strength, muscularity and physical attractiveness that are important to explain the variation in the short-term mating orientation, at least in men, but apparently are not relevant to explain variation in long-term mating orientation [23–26]. In addition, also in men, there are variables such as socioeconomic status that are relevant to long-term but not to short-term mating orientation [23]. These studies suggest that the selective pressures affecting short and long-term mating orientations are partially different and that it is relevant to consider them as separate dimensions.

To summarize, sociosexuality is a highly used construct that has been employed to understand the variability in human mating strategies. In this regard, as we have mentioned before, there are cross-cultural sexual differences in the expression of sociosexuality, in which men are more oriented toward short-term mating than women [6,8]. These differences are in accordance with the typical mating strategies that are expected for men and women according to the parental investment theory [7]. Beyond sexual differences, sociosexuality has also been relevant to investigate the sources of individual variability in mating strategies. In this regard, men expressing traits denoting intrasexual competitive abilities (e.g., muscle mass) or denoting good genes (e.g., physical attractiveness) tend to pursue a short-term mating orientation (or unrestricted sociosexuality) and have a higher number of sexual partners compared to individuals that do not express or express in less degree these traits but do not affect the expression of a long-term mating orientation [23–28]. From an evolutionary perspective, this can be explained since men who are able to compete for mates and are attractive to women would maximize their reproductive success by mating with multiple partners [11]. Accordingly, a positive association between short-term mating orientation and sociosexual behavior with the level of intrasexual competitiveness expressed by men and their physical attractiveness can be expected. For women, the relationship between physical attractiveness and sociosexuality seems to be more complex. Attractive women report a higher short-term mating orientation [29,30] and number of sexual partners and especially long-term partners [28,31]. These results suggest that attractive women have higher mating success but through investing in long-term mating. Also, long-term mating orientation is expected to be highly related to the individual interest in having children and with parenting abilities in both men and women [32]. Finally, the construct of sociosexuality is also related to other variables as sexual orientation, marital status, age of individuals or personality traits among others [6,8,17,33,34].

Considering the relevance of the sociosexuality measure in studies of human mating and the importance of considering short and long-term mating strategies as two different dimensions, this paper aims to provide evidence of the validity of the multidimensional inventory of sociosexual orientation (SOI-M) [12] in Spanish, considering Chilean samples. Although cross-cultural studies have shown good reliability and evidence of validity, it is important to get evidence that supports its use in local samples, especially considering that this version of the instrument has no evidence of validity for use in the South American population. We also believe that this instrument better captures human sociosexuality since it considers strategic pluralism in its way of understanding this construct. For this, sociosexuality has been measured with this instrument in young people who have participated in previous studies since 2016. For the analysis process, the reliability and validity of this scale were evaluated. Physical attractiveness, intrasexual competition scale, and parenting subscale from the components of the mate value survey were employed to assess convergent validity, and exploratory and confirmatory factor analysis was used to assess internal consistency.

## Methods

### Participants

The total sample consisted of 890 subjects; however, we eliminated 22 of them who did not completely answer the questionnaires and 3 who responded with very extreme values to the question about the number of sexual partners during lifetime, which means that they responded more than 100 sexual partners in all their life indicating an unreliable data.

The final sample consisted of 865 subjects (247 women and 616 men) between 18 and 55 years old (M = 23.56; SD = 5.52). Data were collected at the Laboratorio de Comportamiento Animal y Humano between 2016 and 2023 in five different studies and six different samples (study 1: 15th of March 2016–3rd of November 2016; study 2: 17th of November 2017–17th July 2018; study 3: 16th of April 2019–23rd of December 2019; study 4: 20th of November 2020–18th January 2023; study 5: sample 1 from 1st of September 2022–23rd November 2022 and sample 2 from 22nd of May 2023–23rd of August 2023). We took one of them (study 5 sample 1) to perform the exploratory analysis with an independent sample (N = 172). With the other samples, we build the data base for the confirmatory analysis and the ESEM (N = 693).

Recruitment was conducted through the dissemination of announcements in public places, social networks, and databases from previous studies. We did not include data from COVID-19 period because pandemic affected the expression of sociosexual behavior, concretely the number of sexual partners in the last year, due to quarantines (e.g., [35,36]).

## Ethics

This research meets ethical standards for human research. Furthermore, each of the studies from which this database was built has institutional ethical approval by either Universidad de Playa Ancha or Universidad del Desarrollo, and it follows international standards such as the Declaration of Helsinki. The well-being and identity of the participants were protected. Before participating in the studies mentioned above, the participants read and signed a written copy of the informed consent, which explicitly indicated their willingness, the objectives, risks, and benefits of participating, and the possibility of withdrawing from the study at any time. A copy of the document was given to them or sent via email.

## Instruments

**Sociodemographic questionnaire.** This is an instrument with general questions about age, sex, sexual orientation, and relationship status.

**Multidimensional sociosexual orientation inventory.** This is a 20-item questionnaire that measures sociosexuality in attitudinal and behavioral terms. This instrument contains a dimension towards short-term mating orientation (10 items), towards long-term mating orientation (7 items) and contemplates a dimension of previous sexual behavior (3 items). Both the short- and long-term dimensions are built on a 7-point Likert scale and the behavioral dimension is in an open question format. We carry out a direct translation process of the instrument for its application in Chilean Spanish spearkers. The traslation was then evaluated by experts in sociosexuality and psychometrics to adjust some terms. However, we did not perform a back-translation process. In the original study developed by Jackson & Kirkpatrick [12], they obtained Cronbach's α values of 0.95 for the short term; 0.88, for the long term and 0.83 for past behavior.

**Intrasexual competition scale.** This is a 12 items questionnaire that measures the degree to which individuals perceive their interactions with same-sex individuals as competitive, especially in the presence of opposite-sex individuals (e.g., "When I go out, I can't stand it when women/men pay more attention to a same-sex friend of mine than to me"). This questionnaire is answered in 7-point Likert scale (1 = not applicable at all to 7 = completely applicable). We obtained a Cronbach's α value of 0.79 indicating a good reliability but lower than in the original study developed by Buunk & Fisher [37] in which Cronbach's α values of 0.85 and 0.87 were obtained for Netherlands and Canade respectively.

**Parenting.** We employed the parenting factor from the Components of Mate Value Survey that assesses parenting ability and the desire to have children [38]. This factor is composed of 3 items (e.g., "I would make a good parent") answered on a 7-point Likert scale (1 = strongly disagree to 7 = strongly agree). We obtained a Cronbach's α value of 0.71, indicating good reliability. The study of Fisher et al. [38] did not report Cronbach's α values for the factors.

**Self-perceptions of physical attractiveness.** We asked the individuals to rate themselves from 1 (not attractive at all) to 7 (*very attractive*) to the question "How physically attractive do you think you are?

## Data analysis

First, we calculated descriptive statistics for each dimension of the SOI-M and calculated sex differences. Then, in order not to condition the dimensionality to that established in previous studies, we used a sample of 172 participants, men (N = 79) and women (N = 91) between 19–41 years old (M = 21.17; *SD* = 2.31), to perform an exploratory factor analysis, using the weighted least squares estimation method, polychoric correlation matrix, and establishing the dimensionality from the parallel analysis, based on factorial analysis, which has shown substantial advantages over other methods based on eigenvalues [39], through the analysis of the scree plot and the comparison of the eigenvalues of the original extraction

respect to the extraction of the random values simulation. With the remaining sample (N = 693), to establish validity evidence based on the internal structure of the test, we performed an exploratory structural equation model (ESEM) with GEOMIN rotation [40] and Confirmatory Factor Analyses (CFA), both using weighted least squares estimation method (WLSMV), which is robust with non-normal discrete variables [41,42]. Both analyses considered a polychoric correlation matrix, given the ordinal structure of the data [43]. Reliability was estimated for each dimension using Cronbach's alpha and McDonald's omega coefficients, both in non-ordinal versions [44]. Measurement invariance between people of different sexes was assessed using a multigroup CFA (i.e., metric and scalar). Decreases in comparative fit index (CFI) less than .010 were considered evidence of invariance [45]. Model fit was assessed following the recommended cut-points provided by Schreiber [46] for the comparative fit index (CFI), the Tucker-Lewis index (TLI) and the root mean square error of approximation (RMSEA) (i.e., CFI > 0.95; TLI > 0.95; RMSEA < 0.06). Finally, for evidence of validity based on the relations with other variables, we performed Pearson's correlation analysis between the three factors of the SOI-M (short-term mating orientation, long-term mating orientation and past sexual behavior) and the average punctuation in the Intrasexual Competition Scale, the Parenting factor of the Components of Mate Value Survey and the value of the self-perception of physical attractiveness. The Intrasexual Competition Scale and Parenting Factor were only available in a subsample (N = 200) composed only of men. The question about self-perception of attractiveness was only available in another subsample (N = 292) composed of both men and women. It should be noted that both the parallel analysis, reliability coefficients, homogeneity index and Pearson's correlations were obtained using the JASP 0.19.3.0 software, while the ESEM was performed using the Mplus program version 8.2 [47].

## Results

Table 1 shows the sample's mean and standard deviation (in parentheses) concerning each of the dimensions of the SOI-M for each sex and sample. It also contains the t-test for sexual differences. First, for the exploratory factor analysis sample, we found that men (N = 79) reported higher short-term mating orientation and sociosexual behavior than women (N = 91). Women reported higher scores of long-term mating orientation than men, but this difference only reached a marginal level of significance. And second, for the ESEM and confirmatory analyses sample, we found differences in STMO and LTMO between men (N = 537) and women (N = 156). Men reported higher scores of STMO than women and women reported higher levels of LTMO than men. We did not find differences in sociosexual behavior.

### SOI-M dimensionality analysis

First, to establish dimensionality, a parallel analysis was performed, which resulted in the extraction of three factors over the extraction of random variables, as shown in the scree plot (see Fig 1).

**Table 1. Descriptive analyses mean and standard deviation (in parentheses) and t-test to exploratory factor analysis and ESEM/CFA samples.**

|  |  | Men | Women | t | p value |
|---|---|---|---|---|---|
| Exploratory factor analysis* | STMO | 4.90 (1.31) | 4.01 (1.54) | −4.02 | <.001 |
|  | LTMO | 5.75 (1.29) | 6.10 (1.05) | 1.88 | .062 |
|  | SOCIOSEXUAL BEHAVIOR | 5.54 (8.82) | 2.04 (3.78) | −3.27 | .001 |
| ESEM/CFA | STMO | 4.51 (1.39) | 3.86 (1.45) | −5.13 | <.001 |
|  | LTMO | 5.36 (1.26) | 5.99 (1.03) | 6.43 | <.001 |
|  | SOCIOSEXUAL BEHAVIOR | 3.60 (4.45) | 3.59 (3.21) | −.005 | .996 |

Note: STMO (short-term mating orientation), LTMO (long-term mating orientation), sexual behavior.

*Two people were excluded because they did not indicate their sex.

## SOI-M Validity evidence based on internal structure

Based on the 3-factor solution, an ESEM analysis was performed with all the items. Subsequently, those items that showed strong cross-loadings (>.30), small factor loadings (<.40), or that showed excessive redundancy with items of the same dimension were debugged iteratively, leading to a debugged version with 15 items. Fit indexes for both models (the original 20-item scale and the debugged 15-item scale) are shown in Table 2 with two-factor strategies (ESEM & CFA). Both models and strategies (ESEM & CFA) show fit indexes in line with the recommended criteria to declare a good explanation of the observed relations [46]. Reliability and factor loadings for the initial and the final model are shown in Table 3. Factorial loadings of the debugged version showed a large effect size (> 0.50), and all cross-loadings are not relevant (> 0.20). Both models (20 and 15 items) showed good reliability (> 0.80) for all dimensions. There is a little difference in McDonald Omega´s with Cronbach's Alphas in the behavior dimension because there is an important difference between factorial loadings. For the final scale in Spanish, see S1 Table.

Table 4 contrasts the invariance of the instrument between sexes of the final version (15 items), showing invariance at the metric and scalar level.

## Evidence of validity based on the relations with other variables

We calculated correlation coefficients between the three dimensions of the SOI-M and the three selected measures to test for evidence of validity of the 15-item scale. First, we found a positive correlation between the Intrasexual Competition Scale and the short-term mating orientation (r = 0.31, t = 4.588, df = 198, p < 0.001) and no correlation with the long-term

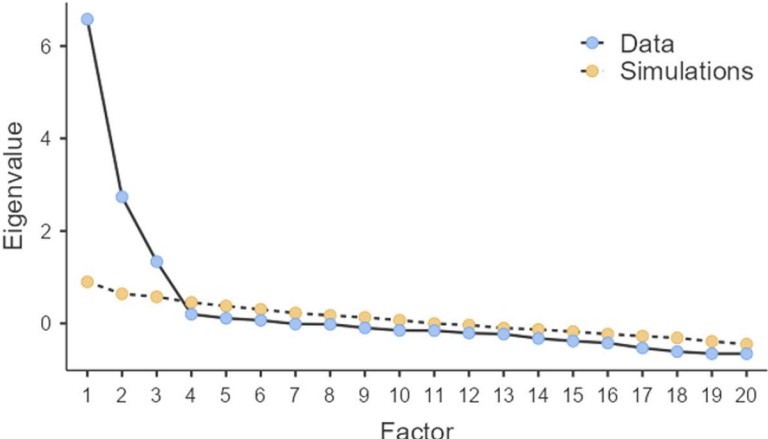

**Fig 1. Parallel analysis to establish factor number.**

**Table 2. ESEM model fit indices.**

|  | N° Par | χ2 | DF | p | CFI | TLI | RMSEA | RMSEA CI 90% | | SRMR |
|---|---|---|---|---|---|---|---|---|---|---|
|  |  |  |  |  |  |  |  | Low | Up |  |
| ESEM (20 items) | 165 | 1222.405 | 133 | <.001 | .955 | .935 | .109 | .103 | .114 | .034 |
| ESEM (15 items) | 120 | 284.782 | 63 | <.001 | .981 | .969 | .071 | .063 | .080 | .021 |
| CFA (20 items) | 131 | 1179.586 | 167 | <.001 | .958 | .952 | .094 | .089 | .099 | .056 |
| CFA (15 items) | 96 | 558.943 | 87 | <.001 | .960 | .952 | .088 | .082 | .096 | .050 |

Note: ESEM and CFA model in both versions of inventory

**Table 3. ESEM results and their factors.**

| Item | ESEM (20 items) | | | ESEM (15 items) | | |
|---|---|---|---|---|---|---|
| | **STMO** | **LTMO** | **Behavior** | **STMO** | **LTMO** | **Behavior** |
| 1. Puedo imaginarme fácilmente estando cómodo/a y disfrutando del sexo casual con diferentes mujeres/hombres. *I can easily imagine myself being comfortable and enjoying "casual" sex with different partners.* | **0.870** | 0.034 | 0.002 | **0.884** | 0.015 | −0.007 |
| 2. Puedo imaginarme disfrutando de un encuentro sexual breve con una mujer/hombre que me resulta muy atractiva/o. *I can imagine myself enjoying a brief sexual encounter with someone I find very attractive.* | **0.881** | 0.123 | 0.024 | **0.908** | 0.111 | 0.005 |
| 3. Puedo imaginarme fácilmente disfrutando de un encuentro sexual de una noche con una mujer/hombre que nunca más vuelva a ver. *I can imagine myself enjoying a brief sexual encounter with someone I find very attractive.* | **0.889** | 0.062 | 0.018 | **0.891** | 0.037 | 0.017 |
| 4. El sexo sin amor está bien. *Sex without love is OK.* | **0.602** | −0.117 | 0.131 | **0.529** | −0.164 | 0.125 |
| 5. Podría disfrutar del sexo con una mujer/hombre que me resulta muy deseable, incluso si esa persona no tiene potencial para una relación a largo plazo. *I could enjoy sex with someone I find highly desirable even if that person does not have long-term potential.* | **0.791** | −0.025 | 0.108 | – | – | – |
| 6. Consideraría tener relaciones sexuales con una extraña/o si se me garantiza que es seguro y la mujer/hombre fuese atractiva/o. *I would consider having sex with a stranger if I could be assured that it was safe and he/she was attractive to me.* | **0.779** | −0.009 | 0.025 | **0.738** | −0.053 | 0.026 |
| 7. Jamás consideraría tener una relación sexual breve con una mujer/hombre. *I would never consider having a brief sexual relationship with someone.* | **0.392** | 0.023 | 0.150 | – | – | – |
| 8. A veces preferiría tener relaciones sexuales con mujeres/hombres que no me importen. *Sometimes I would rather have sex with someone I did not care about.* | **0.711** | −0.090 | −0.080 | – | – | – |
| 9. Creo que debo tomar las oportunidades sexuales cuando las encuentro. *I believe in taking sexual opportunities when I find them.* | **0.711** | −0.143 | −0.119 | **0.650** | −0.179 | −0.060 |
| 10. Tendría que estar profundamente vinculado con una mujer/hombre (emocional y psicológicamente) antes de sentirme cómodo y disfrutando del sexo con ella/él. *I would have to be closely attached to someone (both emotionally and psychologically) before I could feel comfortable and fully enjoy having sex with him or her.* | **0.605** | −0.213 | 0.021 | **0.573** | −0.247 | 0.022 |
| 11. Estoy interesado en mantener una relación romántica a largo plazo con una mujer/hombre especial. *I am interested in maintaining a long-term romantic relationship with someone special.* | −0.114 | **0.812** | 0.091 | −0.007 | **0.892** | 0.017 |
| 12. Espero tener una relación romántica que perdure el resto de mi vida. *I hope to have a romantic relationship that lasts the rest of my life.* | 0.064 | **0.955** | −0.103 | 0.035 | **0.837** | −0.031 |
| 13. Me gustaría tener una relación romántica que dure para siempre. *I would like to have a romantic relationship that lasts forever.* | 0.053 | **0.946** | −0.094 | – | – | – |
| 14. Las relaciones románticas a largo plazo no son para mí. *Long-term romantic relationships are not for me.* | −0.196 | **0.669** | 0.056 | −0.111 | **0.716** | −0.012 |
| 15. Para mí no es importante encontrar una pareja romántica a largo plazo. *Finding a long-term romantic partner is not important to me.* | −0.171 | **0.617** | 0.052 | −0.091 | **0.681** | −0.008 |
| 16. Me puedo imaginar fácilmente involucrándome con una mujer/hombre especial en una relación romántica a largo plazo. *I can easily see myself engaging in a long-term romantic relationship with someone special.* | 0.043 | **0.706** | 0.044 | – | – | – |
| 17. Me puedo imaginar estableciéndome románticamente con una mujer/hombre especial. *I can see myself settling down romantically with one special person.* | 0.052 | **0.752** | 0.120 | 0.118 | **0.742** | 0.019 |
| 18. Durante toda tu vida,¿con cuántas personas distintas has tenido relaciones sexuales? *During your entire life, with how many partners of the opposite sex have you had sexual intercourse?* | −0.004 | −0.010 | **0.922** | −0.044 | 0.003 | **0.985** |
| 19. Durante el último año,¿con cuántas personas distintas has tenido relaciones sexuales completas? *With how many partners of the opposite sex have you had sexual intercourse within the past year?* | 0.132 | −0.051 | **0.565** | 0.114 | −0.051 | **0.573** |
| 20. ¿Con cuántas personas has tenido relaciones sexuales en una sola ocasión? *With how many partners of the opposite sex have you had sex on one and only one occasion?* | 0.063 | −0.004 | **0.712** | 0.067 | 0.024 | **0.703** |

*(Continued)*

**Table 3.** (Continued)

| Item | ESEM (20 items) | | | ESEM (15 items) | | |
|---|---|---|---|---|---|---|
| | STMO | LTMO | Behavior | STMO | LTMO | Behavior |
| McDonald Omega´s h | 0.907 | 0.880 | 0.822 | 0.890 | 0.832 | 0.822 |
| Crobach´s Alphas | 0.905 | 0.875 | 0.683 | 0.888 | 0.825 | 0.683 |

Note: Items in grey were eliminated after ESEM

**p<0.01 *p<0.05

**Table 4. Invariance analysis.**

| | N° Par | χ2 | DF | p | CFI | RMSEA | Δχ2 | ΔDF | PΔχ2 | ΔCFI | ΔRMSEA |
|---|---|---|---|---|---|---|---|---|---|---|---|
| Configural | 96 | 437.566 | 174 | .000 | .943 | .066 | | | | | |
| Metric | 84 | 457.964 | 186 | .000 | .942 | .065 | 20.398 | 12 | .060 | −.001 | −.001 |
| Scalar | 72 | 511.151 | 198 | .000 | .933 | .068 | 73.585 | 24 | .000 | −.010 | .002 |

mating orientation (r=−0.06, t=−0.846, df=198, p=0.393) as expected. However, the Intrasexual Competition Scale was not correlated to the sociosexual behavior (r=−0.03, t=−0.422, df=198, p=0.698). Second, we found a positive correlation between the parenting factor and long-term mating orientation (r=0.43, t=6.702, df=198, p<0.001) and no correlation with short-term mating orientation (r<−0.01, t=−0.141, df=198, p=0.985) and sociosexual behavior (r=−0.06, t=−0.846, df=198, p=0.370) as expected. These results are only applicable to men since the Intrasexual Competition Scale and the Parenting factor were not available in women samples. Finally, for men, we found a positive correlation between self-perception of attractiveness with short-term mating orientation (r=0.18, t=2.775, df=230, p=0.007) and sociosexual behavior (r=0.14, t=2.144, df=230, p=0.035), and no correlation with long-term mating orientation (r=0.06, t=0.912, df=230, p=0.370). For women, we found a positive correlation between self-perception of attractiveness with sociosexual behavior (r=0.26, t=2.033, df=57, p=0.045) but no differences were found either with short-term mating orientation (r=0.10, t=0.759, df=57, p=0.467) or long-term mating orientation (r=0.21, t=1.622, df=57, p=0.103).

## Discussion

Human mating strategies are complex integrated adaptations manifesting in different behavioral tactics that may co-occur simultaneously in short- and long-term mating strategies [9,11]. In this sense, sociosexuality has become a relevant and valid cross-cultural instrument in studying human mating strategies [17,19–21]. However, currently, we lack instruments measuring sociosexuality in the Chilean population with evidence of validity and reliability. The goal of this study was to address this need through the validation of the Multidimensional Sociosexual Orientation Inventory (SOI-M) developed by Jackson and Kirkpatrick [12] since this instrument allows the measurement of sociosexual orientations from a pluralistic view [11]. Unlike other questionnaires, such as the one developed by Penke and Asendorpf [15], the SOI-M includes long-term mating orientation as a separate dimension from short-term mating orientation, which allows us to observe the presence of sequential and simultaneous deployment of mating strategies in both sexes. Results from this study confirmed the three-factor structure found in the original study by Jackson & Kirkpatrick [12]; we found a model with an orientation towards the short-term, the long-term, and sociosexual behavior as factors. However, our results indicate that some items showed small factor loadings (<.40) and excessive redundancy, suggesting a reduced model with a better fit.

In the first place and considering that there was no confirmatory factor analysis in the original study, we conducted an exploratory factor analysis with a subsample of individuals of both sexes to test whether the proposed structure was replicated without any dimensional constraints. Results from the parallel analyses confirmed a three-factor structure. The

model showed indexes fit in line with the recommended criteria to declare a good adjustment to the observed relations [46]. However, in a detailed analysis of the items, it was found that one item showed a small factor loading (<.40) and four items showed excessive redundancies. When these items were removed, the model showed a slightly better fit than the full model, with all factorial loadings above 0.50 and no relevant cross-loadings (<.20) or redundancies. In addition, both models showed good reliability in all dimensions. The reliability of the reduced model was slightly worse than that of the full model. That can be explained by the reduction in the number of items and, especially, by the elimination of redundant items since this redundancy increases the reliability at the cost of overrepresent some content, which can be a problem by overestimating the same dimension by including elements that allude to the same thing but in a different way. In addition, Cronbach's alpha, and McDonald Omega´s showed similar values except for the behavioral dimension. This discrepancy can be explained due to the significant differences in the factor loadings observed for that dimension. When this happens, previous studies have shown that McDonald Omega´s is a more reliable measure of reliability than Cronbach's alpha [48]. Results from the confirmatory analysis also showed that the fit of the reduced model was slightly better than the fit of the full model. In addition, we found metric and scalar invariance when performing the multigroup CFA to the reduced model. This result indicates that the Chilean version of the SOI-M is adequate to test for sexual differences.

Regarding differences by sex, we found significant differences in the three dimensions of the scale, with men scoring higher in the short-term dimension and reporting greater sociosexual behavior but the latter only in the exploratory subsample. In the case of women, the results are also consistent with the evidence in the field, scoring higher in the long-term dimension [1,6,8,12]. That is, men would be showing a less restricted sociosexuality than women, which is consistent with the selective pressures that each sex has had to face during evolutionary history [9,11]. Despite these sexual differences, both men and women could deploy a simultaneous strategy, that is, maintaining a long-term relationship, but without losing and/or seeking opportunities to mate outside the couple. In this sense, future studies should investigate the presence of phenotypes that show simultaneous mating strategies in both sexes. These results support the evidence that sociosexuality should be measured in multidimensional terms and consider the long-term dimension as an independent measure that also includes attitudes and desires around this more committed relationship.

Regarding the relationship between the three factors of the SOI-M with related measures, our results indicate, as expected, that men showing a higher level of intrasexual competitiveness and self-reported physical attractiveness were more oriented toward short-term mating and that men showing higher parental ability and the desire to have children were more oriented toward a long-term mating. In addition, physical attractiveness was also positively associated with sociosexual behavior for men. On the other hand, for women, physical attractiveness was positively related to sociosexual behavior. These results are in accordance with the proposal of the Strategic Pluralism Hypothesis [11] and with the empirical evidence showing that traits denoting attractiveness and intrasexual competitive abilities are associated with a higher expression of a short-term mating orientation and higher number of sexual partners in men [23–28]. However, our results indicate that the level of intrasexual competitiveness was not associated with the number of sexual partners suggesting that men oriented toward short-term mating perceived their interactions with other men as highly competitive, but this not necessarily leads to a higher mating success since this may depend upon other contextual variables and their own physical attractiveness and mate value [12]. In this sense, evidence shows that self-perception of attractiveness does not necessarily correlate with more objective measures, such as the evaluation of attractiveness by third parties [49]. In this study, it was observed that, especially for less attractive people, there is a tendency to overestimate their attractiveness and, consequently, their associated self-reported behavior. Furthermore, it was also observed that they compared themselves to people of similar attractiveness [49], which could contribute to the bias between self-perception and objective measures of attractiveness, such as facial masculinization or feminization [50]. These findings highlight the importance of context in self-assessment of attractiveness and underscore the role of psychological characteristics that are at play. In this sense, it reflects subjective aspects of individuals' self-assessment of their physical attractiveness. For women, attractiveness was

associated with mating success, probably through investments in long-term partners [31,51]. However, this relationship will have to explore in future studies.

One limitation of this study is that, since the research was conducted in different stages and with independent samples, not all data related to other variables were available for analysis, so our samples were heterogeneous in that sense. Furthermore, the translation process was direct from English to Spanish, and we did not perform back translation. Instead, we validated this Spanish version with expert judges.

Overall, our results indicate that both the full model and the reduced model of the Chilean version of the SOI-M show good evidence of validity and reliability, which can be useful for studying human mating strategies in Chilean samples. However, our study also identified five items that have poor psychometric behavior either because have small factor loading or high redundancy. Even though both models show a good fit, there are benefits associated with the use of the reduced model. First, a shorter version of the SOI-M will allow it to be included with other tests at a lower cost associated with participant fatigue. Secondly, reducing the redundancy of the instrument will make it possible to harmonize the representation of content, avoiding the overrepresentation of certain aspects of the instrument. Finally, the short version of the instrument (15 items) would better capture each dimension, resulting in an improvement and more parsimonious measurement of the construct. In conclusion, we verified the psychometric properties of the inventory, which shows that we are measuring the sociosexuality construct as it was described by the authors of the original instrument.

## Supporting information

**S1 Table. Final scale of the Multidimensional Sociosexual Orientation Inventory for Chilean population.** (DOCX)

## Acknowledgments

In memoriam to Daniel Torrico- Bazoberry.

## Author contributions

**Conceptualization:** Oriana Figueroa, Pablo Polo, José Antonio Muñoz-Reyes.

**Data curation:** Oriana Figueroa, Pablo Polo, Gabriela Fajardo, Daniel Torrico-Bazoberry.

**Formal analysis:** Pablo Polo, Rodrigo Ferrer.

**Methodology:** Oriana Figueroa, Pablo Polo, Rodrigo Ferrer.

**Supervision:** Pablo Polo.

**Visualization:** Oriana Figueroa.

**Writing – original draft:** Oriana Figueroa, Pablo Polo, José Antonio Muñoz-Reyes, Gabriela Fajardo, Daniel Torrico-Bazoberry.

**Writing – review & editing:** Oriana Figueroa, Pablo Polo, José Antonio Muñoz-Reyes, Rodrigo Ferrer, Gabriela Fajardo.

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
