## [Decision Letter · Decision Letter 0]

17 Apr 2025

PONE-D-24-46988

Validation of the Multidimensional Social-Sexual Orientation Inventory (SOI-M) in the Chilean population

PLOS ONE

Dear Dr. Polo, 

Thank you for submitting your manuscript to PLOS ONE. After careful consideration, we believe that it has merit but does not fully meet the publication criteria of PLOS ONE in its current form. Therefore, we invite you to submit a revised version of the manuscript that addresses the points raised during the review process. 

Please submit your revised manuscript by May 24 2025 11:59PM. If you need more time to complete the revisions, reply to this message or contact the journal office at plosone@plos.org. When you are ready to submit the revision, log in to https://www.editorialmanager.com/pone/ and select the ‘Submissions Needing Revision’ folder to locate the manuscript file.

When you submit your revised manuscript, please include the following items:

A rebuttal letter that addresses each point raised by the academic reviewer and reviewers. You must upload this letter as a separate file labelled ‘Response to Reviewers’.A marked-up copy of the manuscript that highlights the changes made to the original version. You must upload this file as a separate file labelled ‘Revised Manuscript with Revisions’.An unmarked version of the revised document without revisions. You should upload this as a separate file labelled ‘Manuscript’.

If you wish to make changes to your financial statement, please include the updated statement in the cover letter. Guidelines for the resubmission of figure files are available under the reviewers' comments at the end of this letter.

If applicable, we encourage you to deposit your lab protocols on protocols.io to improve the reproducibility of your results. Protocols.io assigns your protocol a unique identifier (DOI) so that it can be cited independently in the future. For instructions, see: https://journals.plos.org/plosone/s/submission-guidelines#loc-laboratory-protocols. In addition, PLOS ONE offers the option of publishing peer-reviewed articles on laboratory protocols, describing protocols hosted on protocols.io. For more information on sharing protocols, please see https://plos.org/protocols?utm_medium=editorial-email&utm_source=authorletters&utm_campaign=protocols.

We look forward to receiving your revised manuscript.

Best regards,

Vincenzo Auriemma

Academic Editor

PLOS ONE

“This work was funded by FONDECYT postdoctorado project (3220233) from Agencia Nacional de Investigación y Desarrollo, ANID, Chilean Government to OF.”

3. Please note that funding information should not appear in the Acknowledgments section or other areas of your manuscript. We will only publish funding information present in the Funding Statement section of the online submission form. Please remove any funding-related text from the manuscript. 

4. Please note that your Data Availability Statement is currently missing the repository name. If your manuscript is accepted for publication, you will be asked to provide these details on a very short timeline. We therefore suggest that you provide this information now, though we will not hold up the peer review process if you are unable.

**Additional editor comments:**

Dear author, thank you for your patience during the review process. In view of the reviewers' comments, the decision is to request major revisions. Please read the requests that have been made.

**Comments from PLOS Editorial Office:**

We note that one or more reviewers has recommended that you cite specific previously published works. As always, we recommend that you please review and evaluate the requested works to determine whether they are relevant and should be cited. It is not a requirement to cite these works. We appreciate your attention to this request.

[Note: the HTML code is below. Please do not modify it.]

Reviewers' comments:

Reviewer's answers to questions

**Comments to the author**

1. Is the manuscript technically sound and do the data support the conclusions?

The manuscript should describe technically sound scientific research, with data that support the conclusions. The experiments should have been conducted rigorously, with appropriate controls, replications and sample sizes. Conclusions should be drawn appropriately based on the data presented.

Reviewer 1: Yes

Reviewer 2: Yes

2. Was the statistical analysis performed appropriately and rigorously?

Reviewer 1: Yes

Reviewer 2: No

3. Did the authors make all the data underlying the results in their manuscript available?

The PLOS Data Policy requires authors to make all data underlying the results described in their manuscript openly available with few exceptions (refer to the Data Availability Statement in the manuscript PDF file). Data should be provided either as part of the manuscript or supporting information or deposited in a public repository. For example, in addition to summary statistics, the data points underlying the means, medians, and measures of variance should be available. If there are restrictions on the public sharing of data, for example for participant privacy or the use of third-party data, these should be specified.

Reviewer 1: Yes

Reviewer 2: Yes

4. Is the manuscript presented in an understandable manner and written in standard English?

PLOS ONE does not proofread accepted manuscripts, so the language of submitted articles must be clear, correct, and unambiguous. Any typographical or grammatical errors should be corrected during the review process, so please indicate any specific errors here.

Reviewer 1: Yes

Reviewer 2: Yes

5. Review comments to the author

Use the space provided to explain your answers to the previous questions. You may also include additional comments for the author, including concerns related to double publication, research ethics or publication ethics. (Upload your review as an attachment if it exceeds 20,000 characters)

Reviewer 1: I would like to thank the authors for giving me the opportunity to review this work, which I found very interesting and intriguing. The research is well structured and addresses an important topic: the validation of the Sociosexual Orientation Inventory-Multidimensional (SOI-M) in the Chilean population. The strengths of the study are numerous, particularly the methodological approach, which combines exploratory and confirmatory analyses, making the results solid and well supported by evidence.

Strengths:

Robust methodological approach: data analysis with a significant sample (865 participants) and the use of advanced statistical methods such as exploratory factor analysis (EFA), exploratory structural equation modelling (ESEM) and confirmatory factor analysis (CFA) are appropriate choices for testing the validity of the model.

Convergent and Divergent Validity: The results confirm that the SOI-M inventory effectively measures sociosexuality, with strong evidence of both convergent and divergent validity.

Reasoning behind the model reduction: the decision to reduce the initial model from 20 to 15 items by eliminating redundancy and improving the model's fit was well justified, leading to a more parsimonious version of the inventory.

Sexual invariance: the analysis of metric and scalar invariance confirmed that the reduced model is valid for comparisons between men and women, strengthening the generalisability of the results.

Suggested changes:

Line 13:

Current: ‘We used the Multidimensional Sociosexual Orientation Inventory (SOI-M) initially developed by Jackson and Kirkpatrick (2007) for validation in the Chilean population.’

Suggested change: ’We used the Multidimensional Sociosexual Orientation Inventory (SOI-M) developed by Jackson and Kirkpatrick (2007) for validation in the Chilean population.’

Line 33:

Current: ‘We tested the psychometric properties of the instrument on a final sample of 865 subjects (247 women and 616 men) aged between 18 and 55 (M = 23.56; SD = 5.52).’

Suggested change: ‘We tested the psychometric properties of the instrument on a final sample of 865 subjects (247 women and 616 men) aged between 18 and 55 (M = 23.56; SD = 5.52).’

Line 72:

Current: ‘We tested and verified the convergent and divergent validity of the instrument.’

Suggested change: ‘We tested and verified the convergent and divergent validity of the instrument.’

Line 143:

Current version: ’Finally, we verified the psychometric properties of the inventory, which demonstrates that we are measuring the construct of same-sex attraction as described by the authors of the original instrument.’

Suggested edit: ‘Finally, we verified the psychometric properties of the inventory, confirming that we are measuring the construct of sociosexuality as described by the authors of the original instrument.’

I recommend citing the following article, which could further enrich the discussion on psychometric properties and measurement invariance in different cultural contexts:

Diotaiuti, P., Valente, G., Mancone, S., Grambone, A., & Chirico, A. (2021). Metric Goodness and Measurement Invariance of the Italian Brief Version of Interpersonal Reactivity Index: A Study With Young Adults. Frontiers in Psychology, 12, 773363. https://doi.org/10.3389/fpsyg.2021.773363.

How and where to cite:

I suggest including this citation in the paragraph discussing the validity of psychometric tools, particularly when discussing evidence of convergent and divergent validity. A possible integration could be:

Introduction or discussion: ‘Previous studies, such as those by Diotaiuti et al. (2021), have explored the validity of short versions of interpersonal reactivity instruments, providing a useful framework for understanding metric properties and measurement invariance in different cultural contexts’.

Including this citation will help contextualise the findings within a broader body of research on psychometric testing and measurement invariance.

Reviewer 2: In their manuscript, Oriana Figueroa and colleagues validated the multidimensional inventory of socio-sexual orientation (Jason and Kirkpatrick, 2007) for the Chilean sample. To this end, they translated and administered this instrument to a large sample from Chile and provided evidence of its structural validity through exploratory factor analyses (EFAs), confirmatory factor analyses (CFAs) and exploratory structural equation modelling (ESEM).

Overall, I believe the study is sound and makes a valuable contribution to the literature. However, the authors should address some important issues.

My first concern is about the validity of the measurement content: I am not convinced that long-term pairing orientation, and therefore the items that reflect this construct, is a manifestation of sociosexuality. I see it rather as a construct that relates to OS and should definitely belong to its nomological network, but I am not sure that the search for a long-term relationship is, in itself, a form of sociosexuality that is unlimited as opposed to limited. However, if an instrument, albeit multidimensional, refers to a construct (sociosexuality, in this case), then its subscales would to some extent reflect that latent variable and therefore should be highly correlated. This is not only a theoretical consideration that concerns the substantial evidence in favour of the validity of the instrument, which should therefore be thoroughly discussed. It is also an empirical issue that should be taken into consideration when examining the structural evidence. The authors are not very clear about which models they are testing in their CFAs and ESEMs, but I would certainly like to see a comparison between a one-dimensional model, a three-factor model with correlated factors, and a bifactorial model (as in Ciocca et al., 2024). The latter model would also help to break down the total reliability (total omega) into reliability that depends on the general (hierarchical) factor and on specific factors (group omega). In addition, authors can have a nuanced assessment of how much the one-dimensionality - and therefore the calculation of a total score - is supported by the data through the omega evaluation, following the recommendations of Reise et al. (2013), values of ωh > 0.70, ECV > 0.60 and PUCs > 0.80. Ultimately, it would certainly be useful to add the values of these models together with the standardised loads.

My second concern relates to their overall analytical strategy. It is quite clear to me that they aim to test a measurement model – and perhaps compare it with other theoretically valid models such as those mentioned above – and therefore I would have expected to start with a CFA and, in case of a mismatch, an exploratory strategy would have been justified. The authors seem to justify their decision by noting that no CFA was conducted in the original validation paper. However, it is quite clear to me that they theoretically have three different latent variables in mind (SOTM, LOTM and sociosexual behaviour) that are reflected in specific items. Therefore, a more sensible approach would be to randomly split their combined samples in two, conduct a CFA - and ideally compare more competitive CFA models - and then, in case of misfit, explore the causes of the misfits (e.g., by examining modification indices) and explore other solutions through ESEM (especially to identify and remove cross-loading elements) and, if ESEM fails to achieve good fit, perform EFA to understand if the structure has been identified incorrectly. Then, once you arrive at a new valid model, you can test the model through another CFA on the cross-validation sample.

My third concern is about the way the authors verify the validity of the external tests in terms of convergent and discriminant validity. A better approach would be to do this within a CFA framework and check whether the latent variables, rather than the observed scores, are correlated and to what extent they are. This could allow them to check for discriminant validity as recommended by Ronkko and Cho (2022).

My fourth concern is about the huge heterogeneity of their samples in terms of results. The authors should think harder about why they find huge differences in sex differences in sociosexual behaviour between samples. This is another reason why I strongly advise against splitting the data according to the analysis conducted. Perhaps a better approach would be to merge all the datasets and then randomly split the merged sample, as suggested above.

One last important concern is that the authors often omit some important details (see my point-by-point comments below) that could help the reader assess the quality of their evidence.

Here the authors can find other point-by-point comments:

- Page 6: the authors do not cite Ciocca et al. (2024) for the Italian validation of the SOI-R

- Page 11: the translation process should be described in more detail. How did they handle forward translation? Did they also perform a reverse translation? And how was it handled?

- Page 14, Table 1: There are notable differences between the samples. The most notable is the much wider gender difference in the ESM/CFA sample compared to the EFA sample and, even more striking, and somewhat contrary to the literature, the difference in socio-sexual behaviour.

- Page 15: I would like to see more detail on the determination of the number of factors, and it would be useful to provide more detail. Was it conducted through PCA or PAF? Did they also evaluate the scree plot and eigenvalues? And an evaluation of the simple structure? This procedure would have helped to identify the cross-loafing elements in the EFA.

- Page 23: it is not clear why the relationship between attractiveness and intrasexual competitiveness with SOTM should not be relevant for women as well.

- Page 24: ‘In this sense, the evidence shows that the self-perception of attractiveness is not necessarily correlated with more objective measures, such as the evaluation of attractiveness by third parties (Greitemeyer, 2020)’. I would suggest that the authors explore the presumed underlying mechanism.

Minor points:

- Page 20: The degrees of freedom do not apply to the r coefficient itself, but rather to the statistical test used to evaluate its significance

- . Pearson's r coefficient is a descriptive statistic and the test on r could be conducted either through a t-test (with N-2 dfs) or through a z-test. Therefore, I would report, together with the r coefficient, the t statistic together with the relative dfs).

Page 23: ‘Furthermore, physical attractiveness was also positively associated with sociosexual behaviour. For women, physical attractiveness was positively correlated with sociosexual behaviour’. It is not clear whether this is a repetition or whether the first sentence was intended to refer to men.

References

Ciocca, G., Giorgini, R., Petrocchi, L., Origlia, G., Occhiuto, G., Aversa, A., & Liuzza, M. T. (2024). Psychometric characteristics of the Italian version of the revised Sociosexual Orientation Inventory. Archives of Sexual Behavior, 53(8), 3267–3283. https://doi.org/10.1007/s10508-024-02882-w

Reise, S. P., Bonifay, W. E., & Haviland, M. G. (2013). Scoring and modelling psychological measures in the presence of multidimensionality. Journal of Personality Assessment, 95(2), 129–140.

Ronkko., M., & Cho, E. (2022). An updated guideline for assessing discriminant validity. Organizational Research Methods, 25(1), 6–14

6. PLOS authors have the option to publish the peer review history of their article (what does this mean?). If published, it will include the full peer review and all attached files.

If you choose ‘no’, your identity will remain anonymous, but your review may still be made public.

**Do you want your identity to be public for this peer review?** For information about this choice, including how to revoke your consent, see our Privacy Policy.

Reviewer 1: **Yes: **Pierluigi Diotaiuti

Reviewer 2: **Yes: **Marco Tullio Liuzza

[NOTE: If reviewers submitted comments as attached files, they will be attached to this email and accessible via the submission site. Please log in to your account, locate the manuscript record, and check the ‘View Attachments’ action link. If this link does not appear, there are no files attached.]

During submission review, upload your figure files to the Preflight Analysis and Conversion Engine (PACE) digital diagnostic tool, https://pacev2.apexcovantage.com/. PACE helps ensure that your figures meet PLOS requirements. To use PACE, you must first register as a user. Registration is free. Then, log in and go to the UPLOAD tab, where you will find detailed instructions on how to use the tool. If you have any problems or questions while using PACE, send an email to PLOS at figures@plos.org. Please note that supporting information files do not require this step.

---

## [Author Response · Author response to Decision Letter 1]

13 Jun 2025

Reviewer 1: I would like to thank the authors for giving me the opportunity to review this work, which I found very interesting and intriguing. The research is well structured and addresses an important topic: the validation of the Sociosexual Orientation Inventory-Multidimensional (SOI-M) in the Chilean population. The strengths of the study are numerous, particularly the methodological approach, which combines exploratory and confirmatory analyses, making the results solid and well supported by evidence.

Strengths:

Robust methodological approach: data analysis with a significant sample (865 participants) and the use of advanced statistical methods such as exploratory factor analysis (EFA), exploratory structural equation modelling (ESEM) and confirmatory factor analysis (CFA) are appropriate choices for testing the validity of the model.

Convergent and Divergent Validity: The results confirm that the SOI-M inventory effectively measures sociosexuality, with strong evidence of both convergent and divergent validity.

Reasoning behind the model reduction: the decision to reduce the initial model from 20 to 15 items by eliminating redundancy and improving the model's fit was well justified, leading to a more parsimonious version of the inventory.

Sexual invariance: the analysis of metric and scalar invariance confirmed that the reduced model is valid for comparisons between men and women, strengthening the generalisability of the results.

Suggested changes:

Line 13:

Current: ‘We used the Multidimensional Sociosexual Orientation Inventory (SOI-M) initially developed by Jackson and Kirkpatrick (2007) for validation in the Chilean population.’

Suggested change: ’We used the Multidimensional Sociosexual Orientation Inventory (SOI-M) developed by Jackson and Kirkpatrick (2007) for validation in the Chilean population.’

R: Thanks, we have changed that.

Line 33:

Current: ‘We tested the psychometric properties of the instrument on a final sample of 865 subjects (247 women and 616 men) aged between 18 and 55 (M = 23.56; SD = 5.52).’

Suggested change: ‘We tested the psychometric properties of the instrument on a final sample of 865 subjects (247 women and 616 men) aged between 18 and 55 (M = 23.56; SD = 5.52).’

R: Thanks, we have changed that

Line 72:

Current: ‘We tested and verified the convergent and divergent validity of the instrument.’

Suggested change: ‘We tested and verified the convergent and divergent validity of the instrument.’

R: Sorry, but apparently both versions are the same

Line 143:

Current version: ’Finally, we verified the psychometric properties of the inventory, which demonstrates that we are measuring the construct of same-sex attraction as described by the authors of the original instrument.’

Suggested edit: ‘Finally, we verified the psychometric properties of the inventory, confirming that we are measuring the construct of sociosexuality as described by the authors of the original instrument.’

R: Thanks, we have changed the sentence accordingly.

I recommend citing the following article, which could further enrich the discussion on psychometric properties and measurement invariance in different cultural contexts

Diotaiuti, P., Valente, G., Mancone, S., Grambone, A., & Chirico, A. (2021). Metric Goodness and Measurement Invariance of the Italian Brief Version of Interpersonal Reactivity Index: A Study With Young Adults. Frontiers in Psychology, 12, 773363. https://doi.org/10.3389/fpsyg.2021.773363.

How and where to cite:

I suggest including this citation in the paragraph discussing the validity of psychometric tools, particularly when discussing evidence of convergent and divergent validity. A possible integration could be:

Introduction or discussion: ‘Previous studies, such as those by Diotaiuti et al. (2021), have explored the validity of short versions of interpersonal reactivity instruments, providing a useful framework for understanding metric properties and measurement invariance in different cultural contexts’.

Including this citation will help contextualise the findings within a broader body of research on psychometric testing and measurement invariance.

R: Thank you very much for the suggestion. However, after reviewing the work, we saw that it refers to a different construct, and we believe this could be confusing to the reader. In addition, we were not able to locate a good place to cite this work without affecting the current structure and fluency of the text. So we decided not to include it in the manuscript.

Reviewer 2: In their manuscript, Oriana Figueroa and colleagues validated the multidimensional inventory of socio-sexual orientation (Jason and Kirkpatrick, 2007) for the Chilean sample. To this end, they translated and administered this instrument to a large sample from Chile and provided evidence of its structural validity through exploratory factor analyses (EFAs), confirmatory factor analyses (CFAs) and exploratory structural equation modelling (ESEM).

Overall, I believe the study is sound and makes a valuable contribution to the literature. However, the authors should address some important issues.

My first concern is about the validity of the measurement content: I am not convinced that long-term pairing orientation, and therefore the items that reflect this construct, is a manifestation of sociosexuality. I see it rather as a construct that relates to OS and should definitely belong to its nomological network, but I am not sure that the search for a long-term relationship is, in itself, a form of sociosexuality that is unlimited as opposed to limited. However, if an instrument, albeit multidimensional, refers to a construct (sociosexuality, in this case), then its subscales would to some extent reflect that latent variable and therefore should be highly correlated. This is not only a theoretical consideration that concerns the substantial evidence in favour of the validity of the instrument, which should therefore be thoroughly discussed. It is also an empirical issue that should be taken into consideration when examining the structural evidence. The authors are not very clear about which models they are testing in their CFAs and ESEMs, but I would certainly like to see a comparison between a one-dimensional model, a three-factor model with correlated factors, and a bifactorial model (as in Ciocca et al., 2024). The latter model would also help to break down the total reliability (total omega) into reliability that depends on the general (hierarchical) factor and on specific factors (group omega). In addition, authors can have a nuanced assessment of how much the one-dimensionality - and therefore the calculation of a total score - is supported by the data through the omega evaluation, following the recommendations of Reise et al. (2013), values of ωh > 0.70, ECV > 0.60 and PUCs > 0.80. Ultimately, it would certainly be useful to add the values of these models together with the standardised loads.

R: Thank you for the comment. First, and regarding the conceptualization of sociosexuality, we are not sure about the reasons of why long-term is not a form of sociosexuality. From an evolutionary perspective, sociosexuality is widely employed in studies about human mating strategies to assess the relative investment of individuals in short-term and long-term mating. In this sense, there are theoretical and empirical reasons for considering the attitudes toward a short-term mating or casual sex and attitudes toward a long-term mating or pair-bonded relationships as two independent dimensions (e.g., Gangestad and Simpson, 2000; Buss & Schmitt, 2019). Empirical evidence indicates that traits (and contexts) related to (affecting) short-term mating are not necessarily related to (affecting) long-term mating (e.g., Lukaszewski, 2014; Polo et al., 2024) and the other way around (e.g., Fajardo, et al., 2022; Figueroa et al., 2025). In addition, individual differences in sociosexuality are better explained by a two-dimensional perspective rather than a one-dimensional perspective (Hendrickson et al., 2024). In relation to the methodological part, we are testing a 3-factor model with correlated factors in both CFA and ESEM. In addition, as a part of our analytical strategy, we first performed a EFA to check whether the three-factor structure emerges. As you mention in the following comment, this is the structure theoretically proposed by the authors who designed the instrument. Once we had verified that this three-factor structure did indeed emerge, we carried out the CFA and ESEM and refined the questionnaire. Although the analysis strategy you propose is certainly valid, our work was designed to validate this three-factor proposal, so modifying the analysis strategy would compromise the structure of the manuscript.

Buss, D. M., & Schmitt, D. P. (2019). Mate Preferences and Their Behavioral Manifestations. Annual Review of Psychology, 70(1), 77-110. https://doi.org/10.1146/annurev-psych-010418-103408

Fajardo, G., Polo, P., Muñoz-Reyes, J. A., & Rodríguez-Sickert, C. (2022). Long-Term Mating Orientation in Men: The Role of Socioeconomic Status, Protection Skills, and Parenthood Disposition. Frontiers in Psychology, 13. https://doi.org/10.3389/fpsyg.2022.815819

Figueroa, O., Polo, P., Torrico-Bazoberry, D., Fajardo, G., Rodríguez-Sickert, C., Valenzuela, N., Arenas, A., Pavez, P., Belinchon, M., Valdebenito, G., & Muñoz-Reyes, J. A. (2025). The role of the behavioral immune system in the expression of short and long-term orientation in young Chilean men during the COVID-19 pandemic. BMC Public Health, 25(1), 501. https://doi.org/10.1186/s12889-025-21755-y

Gangestad, S. W., & Simpson, J. A. (2000). The evolution of human mating: Trade-offs and strategic pluralism. Behavioral and Brain Sciences, 23(4), 573-587. https://doi.org/10.1017/S0140525X0000337X

Hendrickson, J. N., Peters, S. D., French, J. E., & Maner, J. K. (2024). Distinct individual differences in motivations for pair-bonding and sexual behavior: Implications for close relationships. Personality and Individual Differences, 230, 112779. https://doi.org/10.1016/j.paid.2024.112779

Lukaszewski, A. W., Larson, C. M., Gildersleeve, K. A., Roney, J. R., & Haselton, M. G. (2014). Condition-dependent calibration of men’s uncommitted mating orientation: evidence from multiple samples. Evolution and Human Behavior, 35(4), 319-326. https://doi.org/10.1016/j.evolhumbehav.2014.03.002

Polo, P., Fajardo, G., Muñoz-Reyes, J. A., Valenzuela, N. T., Belinchón, M., Figueroa, O., Fernández-Martínez, A., Deglín, M., & Pita, M. (2024). The role of exogenous testosterone and social environment on the expression of sociosexuality and status-seeking behaviors in young Chilean men. Hormones and Behavior, 161, 105522. https://doi.org/10.1016/j.yhbeh.2024.105522

My second concern relates to their overall analytical strategy. It is quite clear to me that they aim to test a measurement model – and perhaps compare it with other theoretically valid models such as those mentioned above – and therefore I would have expected to start with a CFA and, in case of a mismatch, an exploratory strategy would have been justified. The authors seem to justify their decision by noting that no CFA was conducted in the original validation paper. However, it is quite clear to me that they theoretically have three different latent variables in mind (SOTM, LOTM and sociosexual behaviour) that are reflected in specific items. Therefore, a more sensible approach would be to randomly split their combined samples in two, conduct a CFA - and ideally compare more competitive CFA models - and then, in case of misfit, explore the causes of the misfits (e.g., by examining modification indices) and explore other solutions through ESEM (especially to identify and remove cross-loading elements) and, if ESEM fails to achieve good fit, perform EFA to understand if the structure has been identified incorrectly. Then, once you arrive at a new valid model, you can test the model through another CFA on the cross-validation sample.

R: Indeed, we conducted the exploratory analysis with a subsample (which was not subsequently included in either the ESEM or the confirmatory analyses) to explore whether we found the three factors proposed by the original authors. To do so, we attempted to maintain gender equivalence. After verifying that these three factors were confirmed, we continued with the remaining confirmatory analyses.

My third concern is about the way the authors verify the validity of the external tests in terms of convergent and discriminant validity. A better approach would be to do this within a CFA framework and check whether the latent variables, rather than the observed scores, are correlated and to what extent they are. This could allow them to check for discriminant validity as recommended by Ronkko and Cho (2022).

R: The instruments we used already have proven validity and were used according to their authors' recommendations. We cannot perform the proposed analysis because, as mentioned above, this study was based on previous projects in which we did not always consider the same variables. We verified the validity of the SOI-M using the instruments available for each study indicated in the manuscript. Therefore, we believe that performing structural equation modeling runs the risk of not correctly estimating the model.

My fourth concern is about the huge heterogeneity of their samples in terms of results. The authors should think harder about why they find huge differences in sex differences in sociosexual behaviour between samples. This is another reason why I strongly advise against splitting the data according to the analysis conducted. Perhaps a better approach would be to merge all the datasets and then randomly split the merged sample, as suggested above.

R: We agree with the heterogeneity of the results found. However, this heterogeneity exists empirically, and this demonstrates that the instrument truly captures the variability in reproductive strategies. We also tested metric and scalar invariance, thereby verifying real differences between groups. Therefore, we can affirm that the instrument measures the differences reported in the literature.

One last important concern is that the authors often omit some important details (see my point-by-point comments below) that could help the reader assess the quality of their evidence.

Here the authors can find other point-by-point comments:

- Page 6: the authors do not cite Ciocca et al. (2024) for the Italian validation of the SOI-R

R: Thanks for noticing this. We have included this work in the current version of the manuscript.

- Page 11: the translation process should be described in more detail. How did they handle forward translation? Did they also perform a reverse translation? And how was it handled?

R: We made a translation into Chilean Spanish, which was then evaluated by experts in sociosexuality and psychometrics to adjust some terms. However, we did not perform a back-translation process. For clarity, we have added this information to the manuscript and recognize it as a limitation.

- Page 14, Table 1: There are notable differences between the samples. The most notable is the much wider gender difference in the ESM/CFA sample compared to the EFA sample and, even more striking, and somewhat contrary to the literature, the difference in socio-sexual behaviour.

R: Yes, we agree that there is some variability between the samples. However, all gender differences are in accordance with the theoretical expectations and empirical evidence, including those on sociosexual behavior (e.g., Jackson & Kirkpatrick, 2007; Edelstein et al., 2011; Rhodes et al., 2005). In any case, we have included this as a limitation of the study. In addition, we found scalar and metric invariance between sexes indicating that sex-differences between samples reflect real differences between men and women.

Edelstein, R. S., Chopik, W. J., & Kean, E. L. (2011). Sociosexuality moderates the association between testosterone and relationship status in men and women. Hormones and Behavior, 60(3), 248-255. https://doi.org/10.1016/j.yhbeh.2011.05.007

Jackson, J. J., & Kirkpatrick, L. A. (2007). The structure and measurement of human mati

---

## [Editor Report · Decision Letter 1]

10 Jul 2025

Validation of the Multidimensional Sociosexual Orientation Inventory (SOI-M) in the Chilean population

PONE-D-24-46988R1

Dear Dr. Polo,

We’re pleased to inform you that your manuscript has been judged scientifically suitable for publication and will be formally accepted for publication once it meets all outstanding technical requirements.

Kind regards,

Vincenzo Auriemma

Academic Editor

PLOS ONE
---

## [Editor Report · Acceptance letter]

PONE-D-24-46988R1

PLOS ONE

Dear Dr. Polo,

I'm pleased to inform you that your manuscript has been deemed suitable for publication in PLOS ONE. Congratulations! Your manuscript is now being handed over to our production team.

Kind regards,

on behalf of

Dr. Vincenzo Auriemma

Academic Editor

PLOS ONE